# Effectiveness of Individual Nutrition Education Compared to Group Education, in Improving Anthropometric and Biochemical Indices among Hypertensive Adults with Excessive Body Weight: A Randomized Controlled Trial

**DOI:** 10.3390/nu11122921

**Published:** 2019-12-02

**Authors:** Danuta Gajewska, Alicja Kucharska, Marcin Kozak, Shahla Wunderlich, Joanna Niegowska

**Affiliations:** 1Department of Dietetics, Faculty of Human Nutrition, Warsaw University of Life Sciences—SGGW (WULS), 159C Nowoursynowska Str, 02-776 Warsaw, Poland; j.niegowska@onet.pl; 2Human Nutrition Department, Warsaw Medical University, 02-776 Warsaw, Poland; alicja.kucharska@wum.edu.pl; 3Department of Botany, Warsaw University of Life Sciences—SGGW (WULS), 159C Nowoursynowska Str, 02-776 Warsaw, Poland; nyggus@gmail.com; 4Department of Nutrition and Food Studies, Montclair State University, 1 Normal Avenue, Montclair, NJ 07043, USA; wunderlichs@montclair.edu

**Keywords:** hypertension, individual nutrition education, group nutrition education, overweight, obesity

## Abstract

Objective: The study aims to compare the effectiveness of individual and group nutrition education methods in improving key anthropometric and biochemical markers in drug-treated, overweight-obese hypertensive adults. Methods: The randomized trial included 170 patients with pharmacologically well-controlled primary hypertension and body mass index (BMI) ≥ 25 kg/m^2^. For six months, the patients received six sessions, either one-to-one individual nutrition education (IE, *n* = 89) or group education (GE, *n* = 81), developed by dietitians. Anthropometric measurements, body composition, and fasting measures of biochemical parameters were obtained at baseline and after six months of intervention. Results: 150 patients completed the nutrition education program. The IE group significantly improved in many parameters compared to the GE group, including weight (*p* < 0.001), waist circumference (*p* < 0.001), BMI (*p* < 0.001), systolic and diastolic blood pressure (BP) (*p* < 0.001), fasting plasma glucose (*p* = 0.011), oral glucose tolerance test (OGGT) (*p* = 0.030), and insulin resistance (homeostatic model assessment of insulin resistance, HOMA-IR) (*p* < 0.001). The groups did not differ in terms of total cholesterol, high-density lipoprotein cholesterol (HDL-C) and low-density lipoprotein cholesterol (LDL-C) concentrations. Conclusion: Individual nutrition education is more effective than group education in terms of improving anthropometric and biochemical indices in overweight-obese hypertensive adults.

## 1. Introduction

Hypertension is a serious public health problem and the leading cause of mortality worldwide. In 2010 over 30% of the adult population in the world suffered from this disease [1]. In 2015, the global prevalence of hypertension among adults was estimated to be around 30–45%, with the highest blood pressure level noted in central and eastern Europe [2,3]. Various factors affect this prevalence, such as income and living standards [1]. According to WHO, high blood pressure leads to over nine million deaths each year [4], and it was estimated that by 2025, the hypertensive population will reach 1.5 billion [3]. 

Modifiable risk factors for elevated blood pressure include unhealthy diet, physical inactivity, and excessive body weight [2]. Clinical and observational studies confirm that early diagnosis and effective management can significantly lower blood pressure and prevent complications therewith [2,5,6,7]. Lifestyle change can help prevent chronic diseases such as hypertension and obesity. It also can minimize the side effects of these chronic diseases if they have already been diagnosed [8]. For patients with hypertension, various nonpharmacological interventions are recommended, including weight loss, healthy diet plan, sodium reduction, potassium supplementation, smoking cessation, moderation in alcohol consumption, and increased physical activity [2,8,9,10,11]. 

Dietary intervention is considered as a good therapeutic step for overweight people with high blood pressure [2,8,11]. Several dietary plans have been recognized as effective dietary strategies to prevent and manage elevated blood pressure, including Dietary Approaches to Stop Hypertension (DASH) diet [12], Optimal Macro-Nutrient Intake to Prevent Heart Disease (OmniHeart) diet [13], Portfolio diet [14], Mediterranean diet [15], and vegetarian diet [16]. However, long-term benefits of such diets are difficult to predict, for two main reasons: First, they can significantly differ from patient to patient, and second, such diets’ effects strongly depend on the patient’s ability to adhere to dietary recommendations [2,8,17]. Most patients can, however, prevent hypertension-related health problems with well-chosen lifestyle changes alone. Scientists agree that this strategy, despite being cost-effective and safe for patients, can also bring considerable therapeutic benefits [11,17]. Interestingly, even if such changes do not bring about direct effects in terms of decreasing blood pressure, they can do it in an indirect way, by improving the patient’s physical and emotional state. This, in turn, might enhance the effects of pharmacological treatment.

Any nonpharmacological therapy will be not as effective if the patient does not have sufficient knowledge about the importance of nutrition. Perhaps the most challenging issue is to help patients not only change their lifestyles but also continue these changes in their daily lives. Various educational strategies have been proposed, but their effectiveness is still not fully recognized [18]. Quite likely, it depends on both cultural and individual factors, so what can work for patients from highly developed countries does not have to work for those from underdeveloped ones; what can work for a patient with the tendency to gain weight does not have to work for one without it. Therefore, a patient-centered educational approach is recommended in the management of hypertensive individuals, a model implying that patients are important players in their own therapies. Without the patient being involved, the risk of his or her therapy failing strongly increases, which is why health professionals try to engage their patients in the decision making [18]. 

The study aims to compare the effects of individual and group nutrition education in terms of improving key anthropometric and biochemical markers in drug-treated overweight/obese hypertensive patients. The null hypothesis is that individual or group nutrition education (both provided by dietitians) do not differ in terms of improving anthropometric and biochemical indices in hypertensive patients with excessive body weight.

## 2. Material and Methods

### 2.1. Subjects

The study comprised 170 nondiabetic patients, 74 men and 76 women, aged between 42 and 75 years, with stage 1 or stage 2 essential hypertension (high blood pressure for which there is no clearly defined etiology). They were recruited from two centers: the outpatients Clinic of Arterial Hypertension, the Department of the Cardinal Stefan Wyszyński Institute of Cardiology in Warsaw; and TELMONT Outpatient Medical Center in Warsaw, Poland. All the participants were overweight or obese (BMI > 25 kg/m^2^), had documented hypertension and were pharmacologically treated before entering the study. Special care was taken to continue all medications with dosages unchanged during the study. Patients with normal weight (BMI < 25 kg/m^2^), non-controlled hypertension, chronic kidney, and liver disease were excluded from the study. 

Out of 190 patients who initially volunteered to take part in nutrition education, 170 were enrolled and 150 completed the study, with 83 (40 men and 43 women) in the IE group and 67 (34 men and 33 women) in the GE group. Those twenty participants gave up for various reasons, such as illness, a lack of time, personal issues, and too long a distance to travel to the education center (Figure 1).

### 2.2. Experimental Design

It was a randomized controlled study including 170 hypertensive patients with excessive body weight (BMI > 25 kg/m^2^). The research used a six-month nutrition education intervention program delivered by qualified dietitians. Before the first meeting, the patients’ basic demographic information was collected using a questionnaire to define the target audience. This information included the income level, educational level, occupation, place of residence, family structure, and lifestyle (alcohol consumption and smoking habits). After enrollment, the patients were randomly (using simple randomization) divided into two groups: individual education (IE, 89 patients) and group education (GE, 81 patients).

The individual education (IE) program consisted of six one-to-one dietary counseling sessions per patient. The visits were provided monthly by the dietitians (DG, AK) and lasted from 55 minutes to one hour. The transtheoretical model was used as a nutrient counseling strategy to facilitate behavior changes [17]. The group education (GE) program consisted of monthly lecture/audiovisual sessions delivered by two dietitians (DG, AK) to groups of eight to ten patients. During the six-month study period, the patients in the GE group attended one individual meeting followed by the series of five group educational courses delivered as lectures (60 minutes each). Group sessions were organized to allow the patients to actively contribute to the therapy.

The two groups shared the thematic content of the nutrition education. The lectures/individual counseling focused on general and detailed nutrition education, including the principles of the DASH diet, food quality, the nutritional values of food groups, sources of energy, the role of dietary fiber, food preparation, portion size, recipe modification, healthy fat, and salt and sugar reduction.

Each patient was given a personalized dietary plan and structured meal plans. Realistic goals were established individually for all the patients. Intervention was based on our experience from previous studies of hypertensive patients [19,20]. All the patients received written nutrition materials concerning diet therapy in hypertension and were recommended dietary interventions based on the DASH diet, adapted to Polish eating habits and food preferences. Time duration and the number of consultations provided to both groups of patients remained the same. The total contact time (intervention intensity) ranged from 340 to 360 minutes. The dietitian consultation time was in total 516 hours for IE and 86 hours for GE. Before entering the study, all patients were encouraged by their physicians to lose weight. None of them, however, had met a dietitian. In Poland, referrals to dietitians are not covered by medical insurance. In addition, none had ever previously participated in a nutrition education program. The patients were not recommended to change their usual physical activities during the experimental period. However, they all received the same recommendations for at least a 30 min walk every day or similar activities.

Anthropometric measurements, body composition, and biochemical analysis were conducted at baseline and after the six-month study period. All the methods were used in accordance with the relevant guidelines and the principles of the Declaration of Helsinki. The Medical Ethics Committee of the National Institute of Cardiology in Warsaw approved the study protocol (no. IK-NP-0021-95/911/05). All the patients received both written and oral information concerning the trial and provided their written consent and agreed to participate in the study.

### 2.3. Anthropometric Measurements

All anthropometric measurements, namely height, body weight, waist, and hip circumferences, were recorded by the same research team, following standardized procedures and using calibrated tools [21]. The patients’ heights and body weights were measured in standing position, wearing light clothing (shirt and t-shirt) and without shoes. Height was measured using a stadiometer, with an accuracy of 0.1 cm. The patients were weighed using a medical scale, with an accuracy to the nearest 0.1 kg. Waist circumference was measured at the narrowest point between the highest point of iliac crest and the lower costal margin, using a non-elastic tape to the nearest 0.5 cm. Abdominal obesity was defined as waist circumference > 94 cm in men and > 80 cm in women [21]. Body mass index (BMI), calculated as the patient’s weight (kg) divided by the square of height (m^2^), was used to define categories of body weight. According to the WHO classification [21], overweight and obesity were defined as a BMI ≥ 25 kg/m^2^ and ≥ 30 kg/m^2^, respectively. Body fat mass (FM) was determined by bioelectrical impedance method (BIA) using the BIA 101S, AKERN-RJL bioanalyzer (Italy) device, using the variable frequency current (5, 50, and 100 kHz) [22,23]. The measurement was performed after a night’s rest, on an empty stomach. The participants were asked to reduce their physical activity on the day preceding the measurements, to decrease a potential loss of systemic water.

### 2.4. Blood Pressure

Systolic blood pressure (SBP) and diastolic blood pressure (DBP) were measured by the study physician (cardiologist, JN), using an appropriate arm cuff and a calibrated sphygmomanometer. The BP measurements were carried out between 8:00 and 9:00. The blood pressure was taken after the subject had been sitting upright for at least 5 min [2,8]. The arithmetic mean of two measurements for each subject was calculated.

### 2.5. Biochemical Markers 

Plasma concentrations of glucose, 75 g oral glucose tolerance test (OGTT), serum concentrations of total cholesterol (TC), high-density lipoprotein cholesterol (HDL-C), low-density lipoprotein cholesterol (LDL-C), triglycerides (TG), and plasma insulin concentration were measured using routine techniques. The degree of insulin resistance was estimated by HOMA, according to the methods described by Matthews et al. [24], using the following formula: 

HOMA-IR = fasting serum insulin (µU/mL) × fasting plasma glucose (mmol/L)/22.5.

### 2.6. Statistical Analysis 

The effects of the nutrition education methods on anthropometric and biochemical indices was analyzed using a two-way analysis of variance (ANOVA). The linear model used included two following effects: The main effects of nutrition education and gender and the interaction between them. In the preliminary analysis, the model also included the patients’ ages as a covariate. However, since age affected no variables, the final model did not include this covariate. Before applying the final model, the data were verified for variance homogeneity anthropometric and biochemical indices’ and possible outliers using graphical methods.

The effects of nutrition intervention on anthropometric measurements (body weight, BMI, and waist circumference), BP (systolic and diastolic), and biochemical markers, such as FPG, FIRI, 2 h post-OGTT and HOMA-IR, were visualized by the Tukey mean-difference plot. The plot portrays changes in a variable for each subject that were observed after nutrition intervention; thus, patients with a decrease in the variable’s value are located below the zero line. The x-axis represents a mean value of the variable from the observation made before and after the nutrition education. 

Data collected in the study were analyzed using R software [25]. All variables are expressed as mean ± standard deviation or quartiles. The analyses were conducted using the 0.05 significance level.

## 3. Results

The baseline characteristics of the hypertensive patients who completed the program are shown in Table 1. The mean BMI of the study population was 32.8 ± 4.3 km/m^2^, with the prevalence rate of overweight (BMI ≥ 25.0 km/m^2^) and obesity (BMI ≥ 30.0 km/m^2^) being 26.7% and 73.3%). No significant differences between IE and GE groups were noted in all baseline characteristics.

After 6 months, the follow-up anthropometric variables showed significant changes only in the individually educated (IE) patients (Table 2). Weight loss was found for 81.9% of the IE patients and only 31.3% of the GE patients. Weight gain was observed in 16.2% and 56.7% of the hypertensive patients in IE and GE groups, respectively. The final 6-month average weight change was −3.34 ± 2.89 kg and 0.33 ± 2.93 kg in the IE and GE groups, respectively. This difference was statistically significant (*p* < 0.001). Range of weight loss varied from −0.5 kg to −9.4 kg. At the end of the study, the patients’ weight, waist circumference, and BMI were significantly lower in the IE group (Figure 2). A significant decrease in waist circumference was observed only among the IE patients (105.2 ± 11.5 cm vs 101.2 ± 11.1 cm).

In the IE group, the reduction of anthropometric parameters was accompanied by a significant improvement in insulin resistance (HOMA-IR), blood pressure, and triglyceride control. In the GE group, a significant increase was observed in diastolic BP (83.4 ± 3.8 mmHg vs 85.0 ± 3.7 mmHg, *p* = 0.001), fasting plasma glucose (5.3 ± 0.7 mmol/L vs 5.5 ± 1.1 mmol/L), fasting insulin (14.1 ± 6.4 mmol/L vs 16.4 ± 10.6 mmol/L), 2 h post-OGTT glucose (6.3 ± 2.8 mmol/L vs 7.2 ± 3.1 mmol/L), and insulin resistance (HOMA-IR 3.4 ± 1.7 vs 4.1 ± 2.8) (Figure 3 and Figure 4). The GE patients experienced a small but significant improvement in LDL-C (3.5 ± 0.9 mmol/L vs 3.3 ± 0.8 mmol/L, *p* = 0.012). 

## 4. Discussion

In this randomized controlled study, we compared the effectiveness of two methods of nutrition education in adults with BMI > 25 kg/m^2^ and suffering from essential hypertension. The hypothesis was that nutrition education (both individual and group) delivered by a dietitian do not differ in terms of improving anthropometric and biochemical indices in these patients. Individually educated patients showed significant improvements in more analyzed parameters than did those receiving group education. These findings were rather interesting for those who are involved in medical nutrition therapy, for prevention and management of chronic diseases. Education in group settings is generally considered more cost-effective and labor-efficient, especially for patients suffering from chronic diseases like obesity [26,27]. However, the effectiveness of this counseling method needs to be investigated further. In some studies, on obese patients, group counseling produced significantly greater reduction in weight than individual therapy did [26,27,28,29], and it was recommended as a first-line approach for weight management in young adults [30]. It is, nonetheless, possible that in our study, individual education had a greater impact on individual behavior because it is personalized to a patient’s needs and the dietitian–patient relationship is greater than the one in a group education method. This may-at least partly-explain the large share of the GE patients that failed to complete the study. 

It must be emphasized that the overweight-obese hypertensive patients involved in this study constituted a specific subgroup in a certain phase of behavioral change. They were referred by a physician who previously encouraged them to lose weight. Therefore, they represent a subpopulation of hypertensive patients, ones that have already been informed on the necessity to lose weight by their GPs. Over 82% of Dutch GPs considered weight management of patients as their responsibility [31]. The study from ten European countries revealed that 60% of GPs advised their overweight patients to lose weight. About 58% of physicians felt such an approach was ineffective [32]. The most common obstacles in providing nutrition education by GPs are the lack of time and nutrition knowledge, patient noncompliance, inadequate teaching materials, a lack of counseling skills, a lack of reimbursement, and low confidence of the physician [33]. In an Australian study, GPs identified the cost of referring to a dietitian as the main barrier for patients [34]. Our observations confirm these results as well (data not published). 

A large body of evidence supports the effectiveness of nutrition education in the management of chronic disease, including cardiovascular interventions [35,36] as well as counseling of renal, diabetic, and/or obese patients [37,38,39,40]. Many different models and methods are being used as a strategic approach. There is no single universal method that would work for all patients. In our study, we have identified significant differences between patients subjected to individual and group education, with respect to biochemical parameters such as FPG, FIRI, 2 h post OGTT, and insulin resistance. Hence, individual nutrition education can be more effective in improving clinical conditions of patients with chronic diseases, in effect decreasing mortality and morbidity. Our hypothesis is that, greater weight reduction in patients with IE group could result in better improvement in biochemical parameters. In turn, the lack of the improvement or even deterioration of biochemical parameters in patients from the GE group could have been the result of not following the dietitian’s recommendations. Similarly, De Camargo et al. [41] reported greater decreases in biochemical parameters, such as BP, among individually educated patients than in those receiving group education. The authors stressed the importance of tailored nutrition education and underlined that a strong doctor–patient relationship plays an essential role in achieving the goal of constant blood pressure control. This in turn highlights the central role of interpersonal and subjective aspects in achieving proper medical care.

Accordingly, in our study, the solid dietitian–patient relationship also demonstrated to be more effective in behavioral changes among subjects receiving individual education. These changes reflected in a better and more controllable patients’ behavior and lifestyle changes and could be explained by the significant improvements in the analyzed parameters, both anthropometric and biochemical measurements. Pre- and post-education data suggested that, compared to group education, individual education is more successful in improving patient’s anthropometric and biochemical indices which can result in better hypertension control. Overall, individual nutrition education can thereby reduce the negative health impacts and health care costs of patients with hypertension.

### Limitations of the Study

The study’s limitations include the fact that we did not assess the patients’ readiness to make changes and adherence to recommendations (using validated method). The patients were not divided into adherent or nonadherent groups; random allocation to the study groups, however, should have minimized this effect. In several studies age and gender have been examined as potential predictors of adherence and could impact study outcomes [42,43], however not all studies confirm these results [44]. In addition, most patients suffering from chronic diseases described themselves as highly motivated to change their lifestyles, but they frequently overestimated their real possibility to make such changes. It is also possible that other factors—such as sociological and psychological—influenced the final results. Another limitation of the study is that we did not assess urinary sodium excretion as a potential marker.

Hence, further research is needed to examine the benefits of individual nutrition education in a more general context; such a study would have to cover various sociological, physiological, cultural and environmental factors, and so would have to be conducted on a large scale. So far, several gaps in knowledge regarding the effectiveness of interventions to enhance dietary advice for preventing and managing chronic disease in adult population have been identified [44]. 

## 5. Conclusions

Dietary counseling provided by dietitians can significantly improve the management of patients suffering from hypertension. Between individual and group nutrition education, the former-tailored to a patient’s need-has much greater potential for improving anthropometric and biochemical indices in overweight-obese hypertensive adults. We are reporting the results of a randomized controlled trial, in which all the patients were offered professional counseling in the same setting. As discussed above, the dietitian–patient relationship can be complex, and providing effective counseling is a complex and difficult task. Our study did not cover this topic, but our discussion offers ideas for further studies on the related issues.

This study confirms the potential of nutrition education of adults as an important tool in lowering high blood pressure, a leading risk factor for non-communicable diseases. Our study highlights the need for an intervention which focuses on lifestyle modification in a primary care setting. Further research is needed to study the effectiveness of various approaches to nutrition counseling, one of the key aspects being and the role of dietitians and their skills in multidisciplinary health care teams. 

## Figures and Tables

**Figure 1 nutrients-11-02921-f001:**
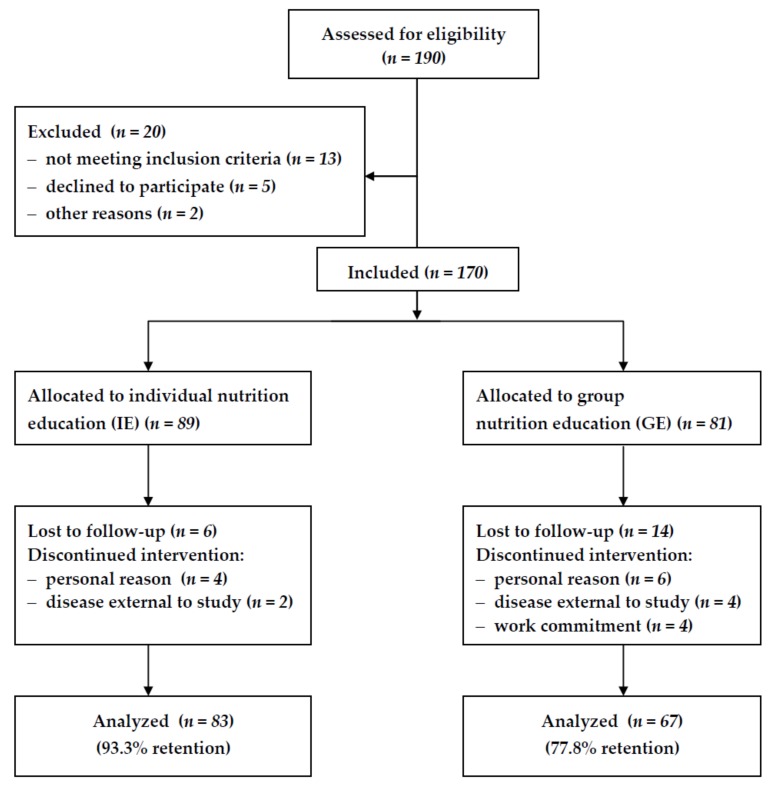
Flow chart of the study.

**Figure 2 nutrients-11-02921-f002:**
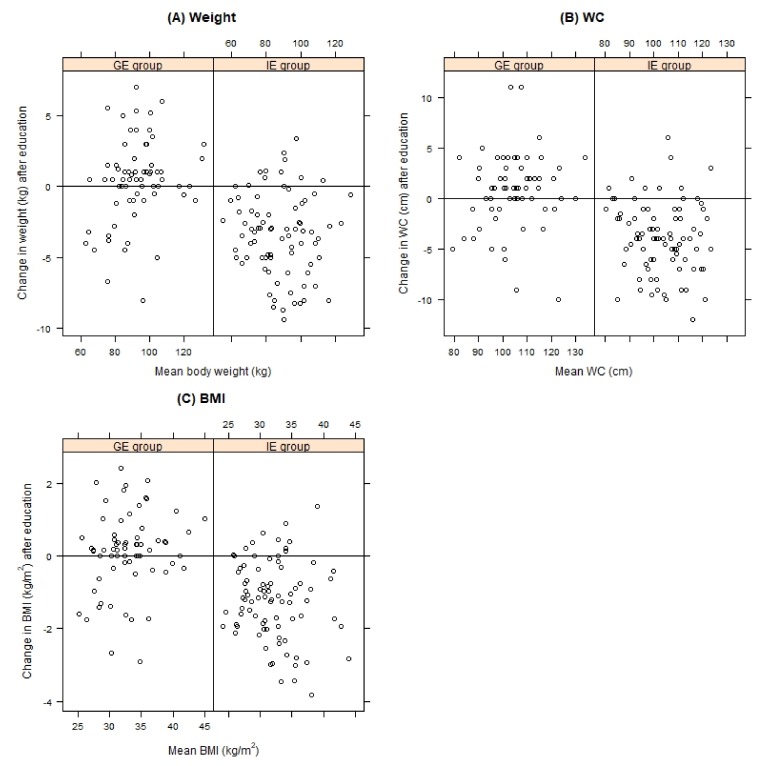
The effect on nutrition education on: (**A**) body weight, (**B**) waist circumferences (WC) and (**C**) body mass index (BMI) for the individually educated (IE group) and group educated (GE group) patients, compared by Tukey mean-difference plot. Points below the zero line indicate subjects with improved parameters after education.

**Figure 3 nutrients-11-02921-f003:**
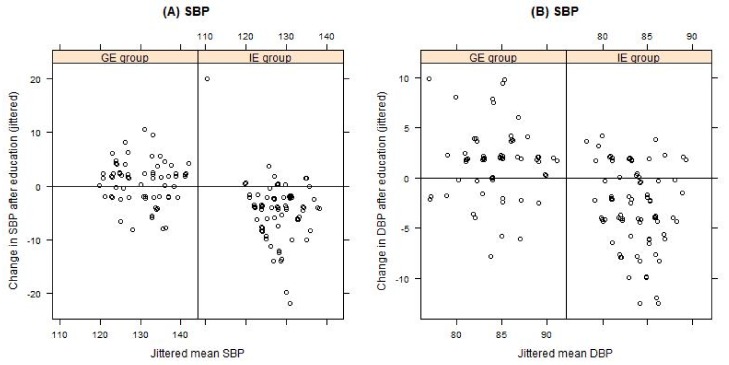
The effect on nutrition education on: (**A**) systolic blood pressure (SBP) and (**B**) diastolic blood pressure (DBP) for the individually educated (IE group) and group educated (GE group) patients, compared by Tukey mean-difference plot. Points below the zero line indicate subjects with improved parameters after education.

**Figure 4 nutrients-11-02921-f004:**
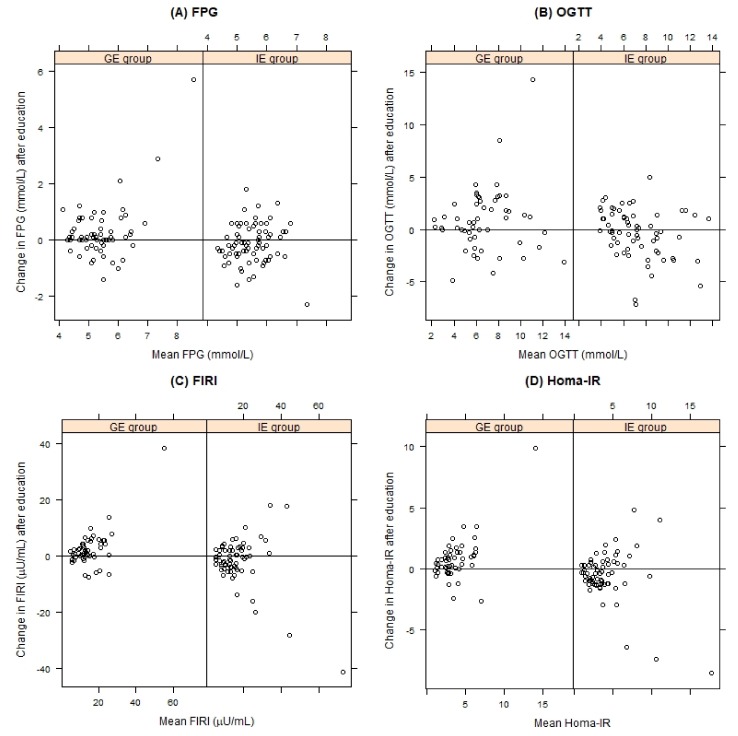
The effect on nutrition education on: (**A**) fasting plasma glucose (FPG), (**B**) 2 h post-OGGTT glucose, (**C**) fasting immunoreactive insulin (FIRI) and (**D**) HOMA-IR for the individually educated (IE group) and group educated (GE group) patients, compared by Tukey mean-difference plot. Points below the zero line indicate subjects with improved parameters after education.

**Table 1 nutrients-11-02921-t001:** Baseline anthropometric and clinical characteristics of patients with essential hypertension who completed the nutrition education program.

Parameter	All (*n*)	Q1	Q2	Q3
Age (years)	60.7 ± 9.3 _(150)_	55	60	68
Weight (kg)	91.0 ± 15.3 _(150)_	80.0	90.0	100.5
WC (cm)	104.7 ± 11.3 _(150)_	97.0	104.0	112.0
BMI (kg/m^2^)	32.8 ± 4.3 _(150)_	29.2	32.3	35.1
Fat mass (%)	35.6 ± 8.9 _(150)_	28.7	35.1	42.0
SBP (mmHg)	130.1 ± 6.3 _(150)_	126.0	130.0	136.0
DBP (mmHg)	84.3 ± 3.7 _(150)_	84.0	82.0	88.0
FPG (mmol/L)	5.4 ± 0.7 _(149)_	5.0	5.5	5.8
FIRI (μU/mL)	15.8 ± 10.1 _(149)_	10.2	13.6	18.7
OGTT (mmol/L)	6.9 ± 3.0 _(141)_	4.8	6.3	9.1
Homa-IR	3.8 ± 2.6 _(150)_	2.3	3.4	4.7
TC (mmol/L)	5.2 ± 1.1 _(146)_	4.4	5.1	5.8
HDL-C (mmol/L)	1.4 ± 0.3 _(145)_	1.2	1.3	1.6
LDL-C (mmol/L)	3.4 ± 0.9 _(145)_	2.7	3.2	3.9
TG (mmol/L)	1.8 ± 0.9 _(146)_	1.1	1.6	2.2

WC = waist circumference, BMI = body mass index, SBP = systolic blood pressure, DBP = diastolic blood pressure, FPG = fasting plasma glucose, FIRI = fasting immunoreactive insulin, OGTT = oral glucose tolerance test, HOMA = IR—homeostasis model of insulin resistance, TC = total cholesterol, HDL-C = high-density lipoprotein cholesterol, LDL-C = low-density lipoprotein cholesterol, TG = triglycerides. Q1 = first (lower) quartile, Q2 = median, Q3 = third (upper) quartile, (the lower and upper quartiles are the smallest values that cut off at least 25% of the values).

**Table 2 nutrients-11-02921-t002:** Changes in anthropometric and biochemical parameters between groups before and after nutrition education. Only those *n* subjects are considered for a parameter for which the parameter’s value is available before and after the dietary intervention.

	Individual Education (IE)			Group Education (GE)			*p* Value ^2^
	Before (*n*)	After	*p* Value ^1^	Before (*n*)	After	*p* Value ^1^	
Weight (kg)	89.7 ± 16.1 _(82)_	86.4 ± 15.9	< 0.001	92.4 ± 14.5 _(66)_	92.8 ± 15.3	0.282	< 0.001
WC (cm)	105.2 ± 11.5 _(82)_	101.2 ± 11.1	< 0.001	104.0 ± 11.1 _(67)_	104.7 ± 11.6	0.139	< 0.001
BMI (kg/m^2^)	32.6 ± 4.5 _(83)_	31.4 ± 4.3	< 0.001	33.0 ± 4.2 _(66)_	33.1 ± 4.4	0.378	< 0.001
Fat mass (%)	34.9 ± 7.6 _(63)_	34.0 ± 8.1	0.069	36.3 ± 10.0 _(65)_	36.4 ± 10.6	0.905	0.100
SBP (mmHg)	130.1 ± 6.2 _(81)_	125.6 ± 5.0	< 0.001	130.3 ± 6.5 _(66)_	130.9 ± 6.2	0.174	< 0.001
DBP (mmHg)	85 ± 3.6 _(81)_	82.1 ± 2.9	< 0.001	83.4 ± 3.8 _(66)_	85.0 ± 3.7	0.001	< 0.001
FPG (mmol/L)	5.5 ± 0.7 _(74)_	5.4 ± 0.7	0.117	5.3 ± 0.7 _(66)_	5.5 ± 1.1	0.034	0.011
FIRI (μU/mL)	17.6 ± 13.1 _(65)_	15.8 ± 10.3	0.082	14.1 ± 6.4 _(50)_	16.4 ± 10.6	0.016	0.005
OGTT (mmol/L)	7.4 ± 3.0 _(64)_	7.1 ± 2.4	0.248	6.3 ± 2.8 _(53)_	7.2 ± 3.1	0.041	0.030
Homa-IR	4.4 ± 3.2 _(65)_	3.8 ± 2.7	0.041	3.4 ± 1.7 _(50)_	4.1 ± 2.8	0.003	< 0.001
TC (mmol/L)	5.1 ± 1.1 _(69)_	5.0 ± 1.0	0.085	5.3 ± 1.0 _(60)_	5.2 ± 0.9	0.150	0.748
HDL-C (mmol/L)	1.4 ± 0.3 _(69)_	1.4 ± 0.3	0.255	1.4 ± 0.3 _(60)_	1.4 ± 0.3	0.151	0.546
LDL-C (mmol/L)	3.3 ± 1.0 _(69)_	3.2 ± 0.9	0.213	3.5 ± 0.9 _(59)_	3.3 ± 0.8	0.012	0.485
TG (mmol/L)	1.9 ± 0.9 _(69)_	1.7 ± 0.8	0.018	1.6 ± 0.8 _(59)_	1.7 ± 0.8	0.703	0.144

^1^—*p*-value within group, ^2^—*p*-value between group for nutrition intervention effect. WC = waist circumference, BMI = body mass index, SBP = systolic blood pressure, DBP = diastolic blood pressure, FPG = fasting plasma glucose, FIRI = fasting immunoreactive insulin, OGTT = oral glucose tolerance test, HOMA = IR—homeostasis model of insulin resistance, TC = total cholesterol, HDL-C = high-density lipoprotein cholesterol, LDL-C = low-density lipoprotein cholesterol, TG = triglycerides.

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
