# Peer review of "Effectiveness of Individual Nutrition Education Compared to Group Education, in Improving Anthropometric and Biochemical Indices among Hypertensive Adults with Excessive Body Weight: A Randomized Controlled Trial"

_nutrients, 2019, doi:10.3390/nu11122921_

Round 1

Reviewer 1 Report

This manuscript is very interesting and inspirational. Authors done very good job. I haven't else recommendation.

Reviewer 2 Report

The Authors have responded to the requests of this reviewer and the text has been modified accordingly

This manuscript is a resubmission of an earlier submission. The following is a list of the peer review reports and author responses from that submission.

Round 1

Reviewer 1 Report

This manuscript is very interesting. Our experiences with individual and group intervention are same. But I think that important fact is if is client employed or if is seniors, mothers who care about small children. For employed client is individual intervention better because the time can be flexible but also this intervention puts larger requirements for motivation and own engagement, activity by client. If is group good it can give very good motivation for each member and it is place for share experiences and sometimes also worry. I think that authors made a very good work and I am happy that I could read it.

Materials and methods are clearly describe.

Results are clearly interpreted and I believe that it will be interesting for readers.Results are clearly interpreted and I believe that it will be interesting for readers.

The limitation of the study should be state after discussion. Their putting in discussion can be confusing.

Reviewer 2 Report

The study analyses the effectiveness of educational programs conducted with either individual or group intervention. This is a topic of great interest in prevention strategies; the study is well conducted; the protocol is suitable and well structured. However, some considerations must be made:

1.It could be useful to know if the adherence to the training path (sessions attended) was different between the two groups and whether this was a determinant of effectiveness of the program. A numerical information of this type, even if not conclusive, could be useful for the planning and the correct sizing of future specific studies. In fact, adherence is often a problem for group courses.

2.I wonder why specify that information on the importance of physical activity was not given. This is an important and often underestimated point.

3.Was the individualized intervention also associated with better adherence to drug therapy?

4.Among the limitations of the study, the lack of an evaluation of urinary sodium should be added.

5.It would be useful to add a comment on the cost analysis for the two groups. The time required for each patient in the individual course is reported in the results section. A comparison with the other group is missing.

Reviewer 3 Report

I have read this paper with a great interest and do have the following comments/suggestions:

Introduction

The authors should consider that a healthy diet plan includes sodium reduction and moderation of alcohol consumption (lines 51 and 52) and these concepts should not be separated.

Statements on lines 69 and 73 deserve to be referenced.

Method

Essential hypertension should be defined. Also it should be clear that participants included in the study have a BMI over 25.

The design of the study (randomized controlled study) should be mentioned in the section on experimental design.

Also, what is the representativeness of the sample size and was the sample size sufficient to detect changes in the parameters under study?

Detailed information on data collected on lifestyle is essential: what types of data were collected and with which method and tool. Similarly how data on basic demographic were collected. It is also important to specify if participants were doing some physical activity and already had some knowledge on DASH diet, and other topics covered by both interventions.

The authors refer to the transtheoretical model as the basic strategy: as so, normally, the readiness of each participant to modify any behavior should have been assessed since it may impact future behavior change. In fact, if the patient is ready to do something (preparation stage), one should also be working not only on knowledge but also on self efficacy to make the change while a focus on decisional balance would be suitable for participants at the stage of pre contemplation and contemplation.

It would also be suitable to further detail the content of the nutrition education provided to  IE and GE patients.

It is mentioned (line 140) that the participant height was assessed to the nearest 0.5 cm. Normally, according to WHO procedures, it should be at the nearest 0.1 cm. Lines 142- 143 deserves to be referenced too (Abdominal obesity was defined as waist circumference > 94...).

In the statistical section, it is unclear why the gender and the age were introduced in the analysis and thereafter, removed.

Table 1: please provide further details on the meaning of Q1 to Q3.

Results

Are all graphs (figures 2 to 4) necessary to present in addition to table 2?

Discussion

The authors come back on the hypothesis which is slightly different than that mentioned in the introduction. I suggest to write the same hypothesis and also, to be more clear about the results. According to table 2, in the GR group, there was no improvement in anthro indices. Moreover, in the IE group, just 3 biochemical indices were improved while several were worst after the intervention in the GE group. Please, further clarify the discussion in light of the actual results.

I am also not sure that one can talk about the effectiveness of interventions in this context. If it is the case, this should be announced and explained in the introduction. In the method section, how effectiveness was assessed should also be described.

I suggest to be careful when stating (lines 255-256) that the cost of referring to a dietitian is a barrier since it has not been assessed in the current study.